# Comparative Transcriptomic Analysis of Root and Leaf Transcript Profiles Reveals the Coordinated Mechanisms in Response to Salinity Stress in Common Vetch

**DOI:** 10.3390/ijms23158477

**Published:** 2022-07-30

**Authors:** Xiaoshan Lin, Qiuxia Wang, Xueyang Min, Wenxian Liu, Zhipeng Liu

**Affiliations:** State Key Laboratory of Grassland Agro-Ecosystems, Key Laboratory of Grassland Livestock Industry Innovation, Ministry of Agriculture and Rural Affairs, Western China Technology Innovation Centre for Grassland Industry, Engineering Research Centre of Grassland Industry of the Ministry of Education, College of Pastoral Agriculture Science and Technology, Lanzhou University, Lanzhou 730000, China; linxsh18@lzu.edu.cn (X.L.); wangqx19@lzu.edu.cn (Q.W.); minxy@yzu.edu.cn (X.M.)

**Keywords:** common vetch, salinity stress, leaf, root, full-length transcripts, yeast

## Abstract

Owing to its strong environmental suitability to adverse abiotic stress conditions, common vetch (*Vicia sativa*) is grown worldwide for both forage and green manure purposes and is an important protein source for human consumption and livestock feed. The germination of common vetch seeds and growth of seedlings are severely affected by salinity stress, and the response of common vetch to salinity stress at the molecular level is still poorly understood. In this study, we report the first comparative transcriptomic analysis of the leaves and roots of common vetch under salinity stress. A total of 6361 differentially expressed genes were identified in leaves and roots. In the roots, the stress response was dominated by genes involved in peroxidase activity. However, the genes in leaves focused mainly on Ca^2+^ transport. Overexpression of six salinity-inducible transcription factors in yeast further confirmed their biological functions in the salinity stress response. Our study provides the most comprehensive transcriptomic analysis of common vetch leaf and root responses to salinity stress. Our findings broaden the knowledge of the common and distinct intrinsic molecular mechanisms within the leaves and roots of common vetch and could help to develop common vetch cultivars with high salinity tolerance.

## 1. Introduction

More than 20% of irrigated land worldwide is hindered by salinity, and nearly 1.5 million hectares are rendered useless each year because of high salinity [1]. Most crop plants are sensitive to salinity stress, which has become a major limiting factor for crop productivity and the sustainable development of agricultural systems. Understanding the molecular mechanisms underlying the response to salinity stress in plants is crucial both for developing crop varieties via genetic breeding methods with high adaptation to salinity stress and for improving productivity on salinized lands.

Generally, salinity stress induces a wide range of physiological, biochemical and molecular alterations in plants, including enhancing the accumulation of oxidizing and reducing substances [2], regulating the levels of reactive oxygen species (ROS) and regulating the expression of genes and transcription factors (TFs) related to salinity stress [3]. In terms of signalling, Na^+^ stress triggers an increase in cytosolic Ca^2+^; thereafter, Ca^2+^-binding proteins further activate downstream pathways [4]. Other secondary messengers linked to Ca^2+^ signalling, such as ROS, are also induced [5]. It has long been known that the ROS level in plants can be induced by salinity stress and plays a dual role in the salinity response: (1) ROS act as signalling molecules mediating salinity tolerance [6], and (2) high levels of ROS also result in oxidative damage and cell death in plants subjected to salinity stress. Thus, plants have evolved complex mechanisms needed to maintain suitable dynamic changes related to ROS production and scavenging. To survive under adverse conditions, plants must perceive and respond to these stresses rapidly by the use of signal transduction pathways mediated by stress hormones such as ethylene, brassinosteroids (BRs), and abscisic acid (ABA) and small molecules such as H_2_O_2_ [7]. For instance, cytoplasmic ethylene insensitive 2 (EIN2) mediates ethylene signalling by imposing translational repression of EIN3-BINDING F-box 1 (EBF1) and EBF2 mRNA, and this protein plays a large role as a receptor of auxin-mediated salinity tolerance in rice [8]. With respect to antioxidative enzyme mechanisms, the chloroplast is the main site of active oxygen generation, and the photosystem I (PSI) and photosystem II (PSII) reaction centres on the thylakoid membranes of chloroplasts in plant cells are the main generators of ROS [9]. Hence, NADH and NAD are products and substrates of ROS scavenging by enzymatic and nonenzymatic (antioxidant) systems. Moreover, flavonoid activation is considered one of the secondary antioxidant systems because of the accumulation of these compounds in response to salinity stress and their contributions to the detoxification of ROS molecules. In addition, high ROS levels not only result in oxidative damage and programmed cell death but also act as signalling molecules [10]. Salinity stress promotes ascorbate efflux via a reduction in apoplastic iron ions and assists in the generation of hydroxyl radicals, thus inducing an increase in intracellular Ca^2+^ in the roots and promoting the generation of ROS signalling [11]. Therefore, the production of ROS acts as an essential component in conferring salinity tolerance to plants.

Given the high efficiency of transcriptomic data collection and new gene discovery, transcriptome sequencing technology has emerged as a revolutionary tool for improving the understanding of the transcriptomic profiles of higher plants under various stresses, especially those whose genome sequence has not yet been determined [12]. Transcriptome technology-based analyses of the molecular mechanisms underlying the response to salinity stress have been conducted in many plant species, including *Arabidopsis* (*Arabidopsis thaliana*) [13], rice (*Oryza sativa*) [14], Petunia (*Petunia hybrida*) [15], Achnatherum (*Achnatherum splendens*) [16], and desert poplar (*Populus pruinosa*) [17]; in these studies, only one tissue type leaf or root tissue was included. However, both common and distinct response mechanisms occur in the leaves and roots when plants encounter salinity stress. Roots are the primary tissue exposed to salinity and are responsible for perceiving stress signals [18]; the effects of salinity stress, which is one of the most harmful factors affecting growth and productivity, are eventually manifested in the leaves, which include a reduced photosynthesis rate and leaf wilting [19]. Transcriptomic data in a previous study indicated that stress-responsive genes are associated mainly with the stress response and photosynthetic and photorespiratory pathways in the leaves, while transcriptional regulation, transmembrane transport, and antioxidant metabolic pathways have been identified in the roots [20]. However, the effects of salinity stress on the transcriptional and organ-specific levels in common vetch remain unknown.

Common vetch is an annual, self-pollinated, diploid leguminous forage species that is highly suited for environments where other food legume or perennial forage legume species are not, especially in cold and dry areas. Common vetch can be grazed [21] or cut for hay production [22], and its seeds can be harvested at relatively low cost and utilized as a valuable feed alternative for animals and humans because of their abundance of digestible protein and minerals [23,24]. In addition, common vetch has been grown as an industrial crop for both methane biofuel and health-promoting food production [25]. A previous study showed that the normal growth and productivity of common vetch can be severely affected by salinity stress [26]. When common vetch seedlings were grown under salinity stress conditions, their height and root length decreased; moreover, the seedlings died when subjected to the greatest level of salinity, and the shoot dry weight, leaf number, and leaf area per seedling decreased up to 80%, 61%, and 89%, respectively [27]. The germination percentage of common vetch seeds and seedling growth also decreased in response to high salinity [28]. Previous studies in common vetch have focused primarily on the characterization and evaluation of the physiological mechanisms that occur in response to salinity stress; however, the detailed mechanism underlying the response to salinity stress of this important legume forage species at the molecular level have not been elucidated. Further identification of the target genes and metabolic processes associated with salinity stress would help to elucidate their regulatory networks and could be applicable to the genetic improvement of common vetch. To gain a deeper understanding of the common and distinct molecular mechanisms in response to salinity stress in common vetch between the roots and leaves, we used a combination of third-generation single molecule real-time (SMRT) isoform sequencing (PacBio) and next-generation sequencing (NGS) technologies to conduct a genome-wide transcriptome analysis of the roots and leaves of plants exposed to salinity stress for different durations. The results of this study are valuable for revealing the coordinated mechanisms regulating the common vetch response to salinity stress. Additionally, the results are useful for functional gene identification and for the development of common vetch varieties with high salinity tolerance.

## 2. Results

### 2.1. Physiological Characteristics of Common Vetch under Salinity Stress

After treated with various salinity stress conditions, the general symptoms of the common vetch seedlings were observed. The seedlings showed signs of wilt beginning at a concentration of 150 mM (Appendix A), and the length of both the shoots and roots decreased significantly at 100 mM NaCl compared with the control (Appendix A), while a significant reduction in biomass occurred at 150 mM (Appendix A). On the basis of the above results, the 150 mM NaCl concentration was chosen for subsequent treatment and transcriptomic studies.

Salinity stress typically adversely affects photosynthesis and leaf MDA contents in plants, leading to an accumulation of energy and consequently resulting in the production of ROS [29]. Electrolyte leakage in salinity-sensitive rice cultivars is greater than that in salinity-tolerant rice cultivars under salinity stress [30]. In the present study, the effects of salinity on photosynthesis, electrolyte leakage, and MDA accumulation in common vetch were analysed. The results showed that the chloroplast content decreased by 2% and 14% at 2 and 24 h, respectively, compared with that of the control (0 h treatment) (Figure 1A). The relative electrolyte leakage content in the roots was greater than that in the leaves, and both increased with prolonged treatment (Figure 1B). Similarly, both the leaf and root MDA contents gradually increased after 2 h and 24 h of treatment, although the increases were not significant (Figure 1C).

### 2.2. Transcriptome Assembly and Annotation of Transcripts under Salinity Stress

To explore the molecular mechanisms of the common vetch response to salinity stress in depth, a pooled sample representing poly(A) RNAs from two tissues (leaves and roots) under different stress treatment durations (0 h, 2 h, and 24 h) was first sequenced to obtain wide coverage of the common vetch transcriptome. After removing the redundant transcripts, the high-quality consensus transcripts of each library were pooled together, a total of 31,393 full-length transcripts were generated from libraries. These sequences ranged from 1–2 kb, 2–3 kb, and 3–6 kb in length (Appendix A). The Illumina HiSeq results were then mapped to the PacBio library, and mapping ratios were calculated, as shown in Table 1. Among them, the Q30 values of the sequences in the 18 libraries reached 90%, and all of the total mapping ratios were greater than 70% (Table 1). In total, 38,826 redundant transcripts were functionally annotated against eight public databases (the Nr, SwissProt, eggNOG, COG, KOG, Pfam, GO, and KEGG databases) (Appendix A). Additionally, 97.78% of transcripts from SMRT were annotated in at least one database. The number of transcripts annotated in the eight databases ranged from 13,755 (34.64%, COG) to 39,411 (99.25%, Nr). Pearson’s correlation coefficients showed that all the correlations among the three replicates ranged from 0.75 to 1, indicating a strong positive correlation and that the sequencing results could be used for further studies (Figure 2).

### 2.3. Identification and Analysis of Genes Differentially Expressed in Response to Salinity Stress

To explore the transcriptomic profiles of common vetch in response to salinity stress, up- or down-regulated transcripts with a fold change ≥ 2 and an FDR < 0.01 between salinity-stress conditions and the control conditions were identified and considered DEGs. A total of 6361 DEGs were identified for at least one treatment time. The number of DEGs in the roots was approximately twice that in the leaves (4442 and 2297 DEGs, respectively). Among them, 1919 and 3364 DEGs were found to be leaf and root-specific, respectively. These results suggested that roots were more sensitive to salinity stress than leaves. Notably, a total of 1078 genes were differentially expressed in both tissues. The comparison of different treatment time points found that only 79 genes (23 up-regulated, 56 down-regulated) were deliberately expressed between L2 vs. L24, versus 854 (534 up-regulated, 320 down-regulated) between L0 vs. L2 and 1116 (393 up-regulated, 723 down-regulated) between L0 vs. L24 in the leaves. Similarly, 451 genes (216 up-regulated, 235 down-regulated) were deliberately expressed between R2 vs. R24, versus 726 (356 up-regulated, 379 down-regulated) between R0 vs. R2 and 1160 (680 up-regulated, 480 down-regulated) between R0 vs. R24 in the roots. These findings indicate that quite a large portion of genes were activated in early stage (Figure 3A). We further analysed the common differentially expressed genes at different treatment time points in leaf, and found that among these 55 genes, 23 were all up-regulated, mainly annotated as zinc finger protein, sodium/calcium exchanger protein, late embryogenesis abundant (LEA) protein, aminotransferase and dehydrin, and the 18 genes that all down-regulated, mainly including methyltransferase and leucine (Appendix A). Meanwhile, among the 140 common differentially expressed genes in root, 58 were all up-regulated and mainly annotated as protein phosphatase 2C (PP2C), NAD dependent epimerase, sugar transporter protein and 2OG-Fe(II) oxygenase superfamily; nine genes that all down-regulated in three comparison groups mainly included helix–loop–helix DNA-binding domain and POT family (Appendix A).

As shown in Figure 3B, more transcripts responded to salinity stress during long-term exposure (24 h) than during short-term exposure (2 h) in both the leaves and the roots. More DEGs were up-regulated in the roots than in the leaves under both short-term and long-term exposure. While in the leaves, the number of transcripts that accumulated in response to salinity stress increased significantly with prolonged stress (Figure 3B).

To obtain an overall picture of the impact of salinity stress on transcriptional profiling, the expression patterns of all DEGs identified in the two tissues at the different time points clustered together. Eight expression patterns were identified in each tissue (*p* < 0.05), including three up-regulated patterns (profiles 4, 6, and 7) and three down-regulated patterns (profiles 0, 1, and 3) (Figure 4A,B). In the leaves, profiles 4, 6, and 7 contained 320, 408, and 387 DEGs, respectively, while profiles 0, 1, and 3 contained 584, 283, and 533 DEGs, respectively. In the roots, profiles 4, 6 and 7 contained 565, 570, and 499 DEGs, respectively, while profiles 0, 1, and 3 contained 392, 470, and 484 DEGs, respectively. Taken together, these results indicate that the expression of many more genes was up-regulated in the roots of common vetch than in the leaves between 0 h and 24 h of stress.

### 2.4. GO Enrichment Analysis

To further determine the coordinated response mechanisms in the leaves and roots of common vetch seedlings under salinity stress, GO category enrichment analysis was applied to determine the function of the DEGs expressed under salinity stress. All sets of leaf and root GO terms are shown in Appendix A. According to the *p*-value and number of DEGs associated with GO terms, the top 20 significantly enriched GO terms were selected and further analysed. These GO terms could be divided into three categories, “Biological process”, “Molecular function”, and “Cellular component”. The category “Biological process”, consisting of 11 functional groups, had the most-significant annotations, the greatest of which was “response to water”, followed by “oxidation-reduction process”, “response to stress”, “integral component of membrane” and “transmembrane transport”, in the leaf tissue. These results indicated that the stress response, antioxidant enzymes, and transmembrane components may be closely associated with salinity stress responses in common vetch. In total, 15, 218, 47, 255, and 86 DEGs were allocated to the “response to water”, “oxidation-reduction process”, “response to stress”, “integral component of membrane”, and “transmembrane transport” categories, respectively. In the roots, “haem binding”, “iron ion binding”, “oxidation-reduction process”, and “extracellular region” were the most enriched terms for the “Molecular function”, “Biological process”, and “Cellular component” categories. Taken together, these findings demonstrate that both iron ions and antioxidant enzymes may play key roles in the roots of common vetch in response to salinity stress (Figure 5).

### 2.5. KEGG Pathway Enrichment Analysis

The DEGs were further annotated to the reference pathways in the KEGG database to explore the key biological pathways in response to salinity stress in common vetch. As shown in Figure 6A, a large number of DEGs in the leaves and roots were assigned to “plant hormone signal transduction” (78 and 135 DEGs, respectively); “starch and sucrose metabolism” (75 and 95 DEGs, respectively); “arginine and proline metabolism” (32 and 51 DEGs, respectively); “pentose and glucuronate interconversions” (23 and 30 DEGs, respectively); “valine, leucine and isoleucine degradation” (20 and 33 DEGs, respectively); “alpha-linolenic acid metabolism” (20 and 43 DEGs, respectively); “fatty acid degradation” (20 and 41 DEGs, respectively); “tyrosine metabolism” (14 and 27 DEGs, respectively); and “ubiquinone and other terpenoid-quinone biosynthesis” (12 and 19 DEGs, respectively), respectively (Figure 6A,B). More DEGs were mapped to these pathways in the roots than in the leaves. These findings suggest that the roots may employ more sensitive and rapid responsive regulatory networks under salinity stress. In addition, flavone metabolism-related pathways were enriched in the leaf and root tissues, for instance, “isoflavonoid biosynthesis” in the leaves and “flavone and flavonol biosynthesis”, “flavonoid biosynthesis”, and “synthesis and degradation of ketone bodies” in the roots. Furthermore, some representative pathways associated with cell wall synthesis, such as “cutin, suberine and wax biosynthesis” in the leaves and “phenylpropanoid biosynthesis” in the roots, were detected. Both tissues had several specific enriched metabolic pathways, for instance, “isoquinoline alkaloid biosynthesis”, “tropane, piperidine and pyridine alkaloid biosynthesis”, and “brassinosteroid biosynthesis” in the leaves and “carotenoid biosynthesis” in the roots (Figure 6A,B).

### 2.6. WGCNA and Identification of Key Genes

To identify the different co-expressed modules in common vetch under salinity stress, we conducted a WGCNA with the 6361 DEGs. The minimum number of genes in each module was set to 30, and a 0.85 threshold was used to merge similar modules. In total, 23 distinct modules (with various colours) were ultimately identified (Figure 7). To identify the modules that were significantly associated with salinity stress, each module was subjected to an interaction analysis based on the *p*-value and the gene numbers from the GO enrichment analysis (Figure 8A,B). The dark grey module contained 100 DEGs, and the purple module contained 96 DEGs (Appendix A). The dark grey and purple modules were seemingly enriched in the response to stress and the response to water. The dark grey and purple module protein interaction networks with scores > 2 were obtained through MCODE (Figure 9A,B). As found in the Pfam annotations, there were also genes annotated as basic region leucine zipper (bZIP), Formin Homology 2 Domain (FH2), biopterin transporter (BT1), late embryogenesis abundant (LEA), Sodium/calcium exchanger (NCX), and dehydrin (DHN) in the dark grey module (Appendix A). The purple module included BT1, the prenyltransferase family (UbiA), the universal stress protein family (USP), and NCX (Appendix A).

### 2.7. Identification of TFs in Response to Salinity Stress

In plants, many TF families function in adaptation to various environmental conditions, including salinity stress [31]. To gain insight into the involvement of TFs and to determine how they cooperatively or differentially respond to salinity stress in common vetch, all the differentially expressed TF-encoding genes were identified. In total, members of 27 and 39 TF families were identified in the leaves (136 members) and roots (256 members), respectively, and the same top 12 families were found in the two tissues (Figure 10). Furthermore, to identify the TFs whose expression is strongly induced in response to salinity stress and explore the basic molecular mechanism of the stress response, the expression levels of 67 common differentially expressed TF-encoding genes between the 24 h salinity-stress treatment and the control treatment in the leaves and roots were compared. The results showed that 13 genes encoding TFs showed opposite expression patterns in the roots and leaves, while the genes encoding 54 TFs had similar expression patterns in both tissues (all up-regulated or down-regulated) (Appendix A).

### 2.8. Validation of Gene Expression by qRT-PCR

To confirm the credibility of our RNA sequencing (RNA-seq) data, the expression profiles of a subset of 13 DEGs involved in oxidation-reduction, membrane components, and hormone signal transduction were selected for qRT-PCR assays. The variation trends and errors of these 13 genes at different treatment points showed a high degree of consistency with the change tendency of the transcript FPKM values (Figure 11A,B). The coefficients of determination obtained by linear regression analysis between the qRT-PCR and transcriptomic data of the leaves and roots were R^2^ = 0.9344 and R^2^ = 0.8466, respectively, and the correlations were positive (Appendix A). The high congruence between the RNA-seq and qRT-PCR results indicated the reliability of the gene expression values in our experiment.

### 2.9. Functional Confirmation of TFs in Transgenic Yeast

To confirm the biological functions of the salinity stress-responsive TFs identified in the two tissues, six TFs that were up-regulated in the two tissues were selected and overexpressed in yeast cells. The results indicated that the growth conditions of the TF-expressing and empty vector control yeast cells were similar under non-stress conditions. However, in the presence of 5 M NaCl, the control yeast cells grew only to a 10^−2^ dilution, whereas the yeast cells expressing each TF gene grew to a 10^−3^ dilution (Figure 12). These results indicated that the *NAC1;2*, *ERF1;2*, and *MYB1;2* TFs can increase the salinity resistance of yeast cells and that they may have strong effects on the salinity stress response in common vetch.

## 3. Discussion

Plants frequently encounter salinity stress in soils and have thus evolved a series of responses and adaptive mechanisms to cope with this abiotic stress. An in-depth understanding of the related gene expression networks and molecular mechanisms underlying plant salinity tolerance would provide valuable information for further salinity-tolerant plant breeding programmes. In this study, we provided a comprehensive transcriptomic profile of the NaCl stress responses of leaves and roots of common vetch for the first time and identified a total of 31,393 full-length transcripts in 18 sample libraries. Of these transcripts, more than 99% were significantly similar (in terms of their sequence) to genes in public databases, the percentage of which was greater than that previously reported in common vetch (Appendix A) [32]. When more stringent criteria of both an FDR < 0.01 and an expression difference greater than twofold were used, a total of 6361 salinity stress-related DEGs after treatment with 150 mM NaCl for 2 h and 24 h compared with the control treatment indicated that these genes directly or indirectly responded to salinity stress in common vetch. On the basis of their expression patterns, all these DEGs clustered into groups exhibiting two general trends: up-regulated or down-regulated after 24 h of salinity treatment (Figure 4). More DEGs were identified in the roots (up-regulated to down-regulated) than in the leaves at 2 h and 24 h of treatment, reflecting the more rapid and severe effects of salinity stress on the roots at the transcriptomic level (Figure 3B). To confirm the reliability of our transcriptome data, we selected 13 DEGs to determine their expression changes via qRT-PCR and further compared with those of RNA-seq data. Of these 13 DEGs, 11 had significant similarity to gene sequences in public databases, two had not and then named “Function unknown protein” (Figure 11). The expression profiles of these 13 selected genes evaluated via qRT-PCR were highly consistent with the transcriptomic results (R^2^ = 0.9344, R^2^ = 0.8466) (Appendix A), indicating that our sequencing data were credible and could be used to identify genes in response to salinity stress. Among these 11 annotated DEGs, some of their orthologs have been characterized to play essential roles in salinity stress. In *Brassica rapa*, the cation exchanger 1 (CAX1) mutations could modify the hormonal balance and resulting in the enhanced salinity tolerance [33]. Under salt stress treatment, the phosphatidylethanolamine-binding protein (PEBP), ABA-induced wheat plasma membrane polypeptide-19 (AWPM-19), and late embryogenesis abundant 4-5 (LEA4-5) were all significantly induced, indicating their essential roles in salt stress [34,35,36]. These results will provide useful clues for further identification of functional genes in response to salinity stress and promote understanding of the molecular mechanisms of salinity tolerance in common vetch.

Among the different mechanisms through which plants adapt to salinity stress, antioxidative pathways and flavonoids play key roles in protecting plant cells from oxidative damage by scavenging free radicals [37,38]. A previous study has shown that oxidation-reduction processes can protect roots from salinity stress in *Populus euphratica* [39]. The oxidation-reduction state of cells is highly important for plant adaptability, normal physiological metabolism, and various biological activities, as well as for signal transduction in response to environmental stimuli. Under the action of various antioxidants, plant cells can maintain oxidation-reduction homeostasis. The main known non-enzymatic antioxidants include superoxide dismutase (*SOD*), catalase (*CAT*), peroxidase (*POX*), cytochrome P450 (*CYP*), and saccharopine dehydrogenase (*SDH*) [3,40,41,42]. Consistently, GO enrichment analysis of the DEGs indicated that some key genes related to oxidation-reduction processes were significantly enriched after salinity treatment in this study (Figure 5). These key genes included those encoding *POX*, *CYP*, *SDH*, and ribulose-1,5-bisphosphate (*RuBP*). The alteration in the activity of these enzymes leads to an increase in NAD(H) binding and then promotes the occurrence of oxidation-reduction processes. Here, we identified two *POX* genes whose transcripts uniformly accumulated in the roots. At the same time, the expression of three genes was inhibited in the leaves, suggesting that *POX*s may function mainly in reducing the production of ROS to protect root cells from damage (Appendix A). NaCl application resulted in a significant increase in the activity of two reductases: cytochrome P450 reductase (CYP) and ferricytochrome b5 oxidoreductase (B5R). Cytochrome b5 can donate reducing equivalents to components involved in a series of lipid-modification reactions, such as desaturation and hydroxylation [43]. Consequently, the oxidation-reduction balance is maintained by these antioxidants through the regulation of reductants and oxidants in a balanced state. Moreover, rapeseed leaf discs subjected to osmotic stress were shown to have up-regulated levels of *LKR/SDH* gene expression, and the activity of SDH was induced by osmotic shock at levels proportional to the intensity of the osmotic treatment [44]. This phenomenon is in agreement with our findings, in which three *CYP*s and three *SDH*s were significantly activated in both tissues (Appendix A). Salinity stress can cause an increase in oxygenase activity in plant leaves while decreasing the carboxylase activity of Rubisco, and proline also plays an important role in this response [45]. To alleviate oxidative damage, the antioxidant defence system in common vetch was significantly activated via ROS detoxification of antioxidants (*POD*s, *CYP*s, *SDH*s, *RuBP*s) (Appendix A). The *RuBP* transcript abundance did not change in the roots but significantly decreased in the leaves, indicating that the leaves employ a rapid response mechanism during the oxidation-reduction process.

Flavonoids perform a variety of physiological and biological functions and act mainly as antioxidants to prevent damage from ROS generated during stress conditions [46]. Relevant research shows that responses of the Antarctic moss *Pohlia nutans* under salinity stress involve activating phytohormone signalling pathways that in turn trigger flavonoids antioxidant defence systems, to protect cells and scavenge ROS [47]. During flavonoids system reaction, Chalcone synthase (CHS) catalyses the first committed step in the biosynthesis of flavonoids in plants. Afterward, chalcone isomerase (CHI) converts naringenin chalcone to flavanone (2S)-naringenin. In plants, naringenin is converted into flavones by flavone synthase I (FNSI), flavone synthase II (FNSII) or CHI, and isoflavone synthase (IFNS) catalyses the conversion of deoxy flavonoids to generate isoflavonoids [48]. According to the report, CHS-box chalcone family gene plays an important role under abiotic stress in *Medicago truncatula*, *MtCHS15* and *MtCHS12* show significant up-regulated under high salt stress [49]. *Pohlia nutans* FNSI also increased the enzyme activities and gene transcription levels of reactive oxygen species (ROS) scavengers, protecting plants against oxidative stress. Moreover, overexpression of *PnFNSI* enhanced the flavone biosynthesis pathway in *Arabidopsis* [50]. In the present study, the KEGG category “isoflavonoid biosynthesis” was significantly enriched in the leaves, and “flavone and flavonol biosynthesis”, “flavone biosynthesis”, and “synthesis and degradation of ketone bodies” were significantly enriched in the roots. Moreover, “isoflavonoid biosynthesis” and “flavone and flavonol biosynthesis” were the most enriched pathways in the leaves and roots, respectively (Figure 6A,B). In total, the expression of 6, 3, 4, and 2 *CHS*, *CHI*, *FNS*, and *IFNS* DEGs was significantly affected (Appendix A). The upregulation of nearly all four types of DEGs suggested that antioxidants and flavonoids were rapidly activated in the roots of plants under salinity stress. However, the DEGs in the leaves were associated mainly with the formation of isoflavones, indicating the specificity of the antioxidant mechanism of isoflavones in the leaves suggested that rate-limiting enzyme in the synthesis pathways of plant flavonoids and isoflavones plays an important role in plant abiotic stress.

Enhanced tolerance to salinity stress has been shown to be associated with abscisic acid (ABA), ethylene, auxin, jasmonates, and BRs [51]. Recent studies have also indicated that photosynthesis is affected by plant hormones in response to different abiotic stress conditions [52]. Consistently, our transcriptome analysis also revealed that hormone signal transduction pathways were activated at the molecular level in response to salinity stress and had the most enriched genes compared with those of the other categories in both tissues (Figure 6A,B). Moreover, previous research showed that the expression of *OsIAA9* and *OsIAA20* in rice significantly increased when plants were subjected to high-salinity conditions [53]. The perception and control of the transmission of auxin signals are mediated by proteins belonging to three families: receptors (F-box proteins), repressors [auxin/indole-3-acetic acids (Aux/IAAs)] and auxin response factor (ARF) transcriptional activators [54]. In this study, the accumulation of transcripts of the Aux/IAAs, ARFs, and F-box proteins was inhibited during the 24 h NaCl treatment in both tissues. However, other genes, including five *PP2C*s and two *ETH*s, were uniformly up-regulated after treatment with NaCl for 24 h (Appendix A). Previously, NaCl was shown to trigger the stabilization of Aux/IAA repressors, leading to the downregulation of auxin signalling upon Arabidopsis NaCl treatment [55]. This is in agreement with our results, indicating that auxin distribution and signalling mediate the common vetch response to salinity stress.

Many TFs, such as ERFs, NACs, MYBs, WRKYs, bZIPs, and bHLHs [41,42,56,57], have been proven to play functions in response to salinity stress in plants. Within the common vetch leaf and root tissue transcriptome, genes encoding 136 and 256 TFs in 27 and 39 TF families, respectively, were identified as being differentially expressed under salinity stress. In addition, the top three TF families whose members were significantly induced under salinity stress in common vetch were the ERF, NAC, and MYB families (Figure 10), and the results from the yeast system indicated that these three TF families exerted marked regulatory effects on improving common vetch salinity stress tolerance (Figure 12). ERFs are widely involved in the regulation of plant growth and response to various stresses. Salinity and drought resistance significantly improved when the *JcERF2* gene of *Jatropha curcas* was overexpressed in tobacco [58]. In common vetch, a total of 12 *ERF* genes were significantly induced in both tissues. Six of these genes were up-regulated in the leaves and roots, and four were down-regulated in the roots but up-regulated in the leaves. These findings indicate that *ERF*s may also play critical roles in activating the expression of downstream genes in response to salinity stress in common vetch. The diverse expression of these DEGs suggested that common vetch may reduce salinity damage by regulating ethylene synthesis and decomposition at the same time. Expression of *Populus euphratica NAC1*, which contains a conserved NAC domain, is strongly induced by drought and salinity stress, and transgenic Arabidopsis overexpressing *PeNAC1* has increased tolerance to salinity stress, suggesting that *PeNAC1* plays a role in the plant response to salinity stress by regulating Na^+^/K^+^ homeostasis [58]. When common vetch is under salinity stress, the most up-regulated TF genes were members of the NAC family. Six of these genes were co-activated and synergistically up-regulated between the leaves and roots. Members of the MYB family are crucial to regulating plant stress resistance, and this TF family has the largest number and most diverse functions related to the plant stress response; moreover, *MYB*s have been proven to be widely involved in the regulation of the flavonoid metabolism pathway. In this study, except for one transcript (referred to as F01.PB8561) that was down-regulated in the leaves, *MYBs* were up-regulated in both the aboveground and belowground tissues. We speculate that *MYBs* participate in the regulation of the amount of photosynthetic pigments by regulating the secondary metabolic process of flavonoids, thus affecting the chlorophyll content of common vetch and the photosynthetic ability of leaves.

## 4. Materials and Methods

### 4.1. Plant Materials and Growth Conditions

Seeds of Lanjian No. 1 common vetch were rinsed and moistened thoroughly in distilled water and 20% sodium hypochlorite, after which they were then placed on two layers of filter paper that were moistened with distilled water in square-shaped Petri dishes and allowed to grow at 22 °C for 4 days. Twenty-five seedlings of uniform length were sown in 60-well plates supported by a plastic container and then hydroponically grown in half-strength Murashige and Skoog (1/2 MS) nutrient solution at a pH of 5.8. The seedlings were grown in a controlled greenhouse under a 16 h light/8 h dark photoperiod, a 22 °C temperature and a photosynthetic photon flux of 180 μmol m^−2^ s^−1^. During normal growth of the seedlings, the nutrient solution was replaced every 3 days, and salinity stress was imposed after 5 days of growth.

To determine appropriate salinity stress concentrations for common vetch, an extensive preliminary experiment on the effects of various NaCl concentrations (50 mM, 100 mM, 150 mM, 200 mM, 250 mM, 300 mM, 350 mM, and 400 mM) on the shoot length, root length, and fresh weight of 6-day-old common vetch seedlings was performed. Healthy plants of the same age growing in normal 1/2 MS nutrient solution were used as controls. Physiological measurements were performed, and samples for RNA isolation were collected from seedlings at 0 h (control conditions), 2 h, and 24 h after NaCl treatment. Each of the above experiments was repeated three times. To reduce circadian rhythm effects, plants under the control and salinity conditions were grown in parallel, and their leaf and root samples were harvested at the same time after 48 h.

### 4.2. Physiological Measurements

All 24 samples (three treatments (150 mM NaCl for 0 h, 2 h, and 24 h) × two tissues (leaves and roots) × four biological replications) were immediately harvested and subjected to physiological measurements. Electrolyte leakage was determined using a DDSJ-308A conductivity meter (Shanghai Instrument Co., Ltd., Shanghai, China), and the last conductivity reading was obtained under constant temperature at 25 °C in a boiling water bath, with leaf and root tissues autoclaved at 100 °C for 30 min. The malondialdehyde (MDA) content was measured by the thiobarbituric acid reaction method [59]. The chlorophyll content was measured using a SPAD-502 portable chlorophyll analyser (Konica Minolta, Inc., Tokyo, Japan) according to the manufacturer’s instructions.

### 4.3. RNA Isolation, Library Construction, and Sequencing

#### 4.3.1. RNA Isolation

A total of 18 samples (three treatments (150 mM NaCl for 0 h, 2 h, and 24 h) × two tissues (leaves and roots) × three biological replications) were collected and used for RNA extraction. Total RNA from these samples was isolated using TRIzol reagent (Invitrogen, Carlsbad, CA, USA). The purity and integrity of the RNA were subsequently assessed via a Nanodrop instrument and a 2100 Bioanalyzer (Agilent Technologies, Palo Alto, CA, USA). Agarose gel electrophoresis was used to detect RNA degradation and contamination. Equal amounts of total RNA from each sample with an RNA integrity number ≥ 7.0 and a 28S/18S ratio ≥ 1.0 were sampled together for PacBio sequencing (Pacific Biosciences, Menlo Park, CA, USA). For Illumina sequencing (Illumina, San Diego, CA, USA), an indexed library of 18 internodal RNA samples was prepared and sequenced. The library comprised 6 samples (in duplicate): L0 (leaf control), R0 (root control), L2 (2 h salinity stress of leaves), R2 (2 h salinity stress of roots), L24 (24 h of salinity stress of leaves), and R24 (24 h of salinity stress of roots).

#### 4.3.2. PacBio Library Construction and Sequencing

Four micrograms of mixed total RNA was used to synthesize first-strand cDNA by a SMARTer^TM^ PCR cDNA Synthesis Kit (Clontech, Mountain View, CA, USA). cDNAs of different sizes were fractionated and selected using BluePippin (Sage Science, Inc., Beverly, MA, USA). A Clontech SMARTer PCR cDNA Synthesis Kit was used to produce Iso-seq libraries and generate three libraries with sequences that were 1–2, 2–3, and 3–6 kb in size. After size selection, another amplification was performed to obtain full-length cDNAs. A SMRT dumbbell was connected and used for exonuclease digestion. Finally, BluePippin was used for secondary screening to generate the sequencing library.

#### 4.3.3. Illumina HiSeq Library Construction and Sequencing

Poly(A)-containing mRNA molecules were purified from 1 μg of RNA per sample using poly-T oligomagnetic beads. The extracted mRNAs were fragmented into pieces randomly using RNA fragmentation buffer. The cleaved Illumina sequencing mRNA fragments were reverse-transcribed into first-strand cDNAs using random hexamers. Second-strand cDNA was subsequently synthesized via DNA polymerase I and RNase H and further purified by AMPure XP beads. After phosphorylation of the 5′ end and sticky “A” bases at the 3′ end, the cDNA fragments were ligated to adapters and used for amplification. cDNA libraries were ultimately sequenced on an Illumina HiSeq high-throughput sequencing platform. Afterward, the Q30 value (number of errors per 1000 bases) and GC content were calculated to evaluate the quality of the clean data. The clean reads were mapped back to the reference transcript sequences via Bowtie2 [60], and the read count of each transcript was determined from the mapping results.

#### 4.3.4. Mapping Reads from the PacBio Library and Annotation Analysis

After quality control checks were sequenced, low-quality conforming sequences were corrected on the basis of the Illumina HiSeq high-throughput sequencing results. Clean reads were separated from the raw data by removing adapter sequences and low-quality reads. The high-quality clean reads were then used as a reference for further transcriptomic data analysis. The PacBio RSII data were applied to modify the reference genome by the use of Bowtie (v2.2.3). The gene expression levels were quantified by the RSEM software package [61] and normalized by the fragments per kilobase of transcript per million mapped reads (FPKM) method [62]. All expressed transcript functions were annotated to eight public databases by BLAST software (v2.2.26): the NCBI non-redundant (Nr) protein, Protein Family (Pfam), SwissProt (which contains manually annotated and reviewed protein sequences), EuKaryotic Orthologous Groups (KOG), Clusters of Orthologous Groups of proteins (COG), evolutionary genealogy of genes: Non-supervised Orthologous Groups (eggNOG)), Gene Ontology (GO), and Kyoto Encyclopedia of Genes and Genomes (KEGG) databases.

### 4.4. Identification of Differentially Expressed Genes (DEGs)

The expression level of transcripts was calculated by quantifying the reads according to the FPKM values [63]. The transcript fold change was calculated by the formula of log2 (FPKM treatment/FPKM control) using an MA plot-based method with a random sampling model via the R package DEGseq [64]. DEGs were determined on the basis of a |fold change ≥ 2| and a false discovery rate (FDR) < 0.01. Cluster analysis was performed, and DEG expression patterns were assessed using the Biomarker (BMK) Cloud platform (http://www.biocloud.net/ accessed on 5 March 2021). The expression level (FPKM) of mutually expressed genes among the different treatments was analysed using TBtools [65]. Venn diagrams were generated using the Venny 2.1 tool (https://bioinfogp.cnb.csic.es/tools/venny/index.html accessed on 3 March 2021). GO and KEGG pathway enrichment analyses of the DEGs were performed via the BMK online platform. Weighted correlation network analysis (WGCNA) was conducted by R software [66]. Expression correlation coefficients were calculated for gene networks via the scale-free topology model [67]. To identify biologically significant modules, modules were used for GO enrichment analysis. The resulting networks were visualized with Cytoscape v3.7.1 and MCODE [68]. Potential TFs were identified by the PlantTFDB, with the default parameters (http://planttfdb.gao-lab.org/blast.php accessed on 16 March 2021).

### 4.5. qRT-PCR Analysis

cDNA was synthesized from 500 ng of total RNA using ReverTra Ace^®^ qPCR RT Master Mix (with DNase) (Toyobo Biotech Co., Ltd., Shanghai, China) according to the manufacturer’s instructions. Thirteen gene-specific primers were used and designed via a tool on the NCBI website (https://www.ncbi.nlm.nih.gov/tools/primer-blast/ accessed on 5 November 2020). The sequences are shown in Appendix A. qRT-PCR analysis was performed on a CFX 96 Real-Time PCR system (Bio-Rad) using 2xSG Fast qPCR Master Mix (Sangon Biotech, Shanghai, China). Three replications were included per sample. The thermal cycling conditions were as follows: 95 °C for 3 min, followed by 40 cycles of 95 °C for 3 s and 60 °C for 30 s. The relative gene expression levels were calculated according to the 2^−ΔΔCT^ method [69].

### 4.6. Salinity Tolerance Tests of Transgenic Yeast

Six up-regulated TFs that were significantly enriched in both leaf and root tissues were chosen to determine their potential function in yeast. The specific primers used for gene cloning are shown in Appendix A. The complete open reading frames (ORFs) of these TFs were cloned into a pYES2 vector (which contained a URA3 selection marker driven by the GAL1 promoter). The resulting vectors were named pYES2:*NAC1* and pYES2:*NAC2*; pYES2:*ERF1* and pYES2:*ERF2*; and pYES2:*MYB1* and pYES2:*MYB2*. The plasmids containing target genes and the empty pYES2 vector were subsequently transformed into INVSc1 yeast strains via the lithium acetate method [70]. After treatment, the yeast cells were spotted onto SC-Ura media in 2 mL aliquots of 10-fold serial dilutions (1, 10^−1^, 10^−2^, 10^−3^, 10^−4^, 10^−5^, and 10^−6^). Afterward, the cells were incubated at 30 °C to monitor their growth for 36 h.

## 5. Conclusions

By combining RNA-seq and SMRT sequencing, this study provides an overview of the cooperative changes in transcript abundance in the leaves and roots under salinity conditions. Our results shed light on the general mechanisms of the stress response and adaptations at the molecular level in common vetch. DEGs responsive to salinity suggest that different regulatory patterns occur in the leaves and roots. These DEGs overlapped or diverged in the cascades of molecular networks involved in oxidation-reduction, hormone signal transduction, flavonoid metabolism, and functional TFs. Compared with the aboveground tissues, the belowground tissues responded more rapidly and were more sensitive to salinity stress in common vetch. Taken together, our results provide a crucial view for future efforts to understand the biochemical and molecular mechanisms underlying the response to salinity stress in common vetch. Moreover, the functional genes involved have the potential to be used for the development of novel common vetch varieties with improved productivity and stress tolerance.

## Figures and Tables

**Figure 1 ijms-23-08477-f001:**
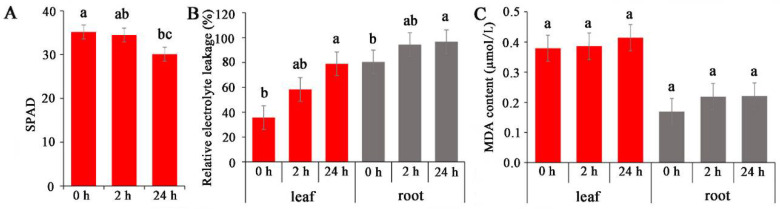
Analysis of dynamic physiological effects under NaCl stress. (**A**) Chlorophyll content. (**B**) Electrolyte leakage. (**C**) MDA (Malonydialdehyde) content. The results are the means and SDs of three replicates. The different letters above the bars indicate significant differences at the 0.05 level according to Duncan’s multiple range test.

**Figure 2 ijms-23-08477-f002:**
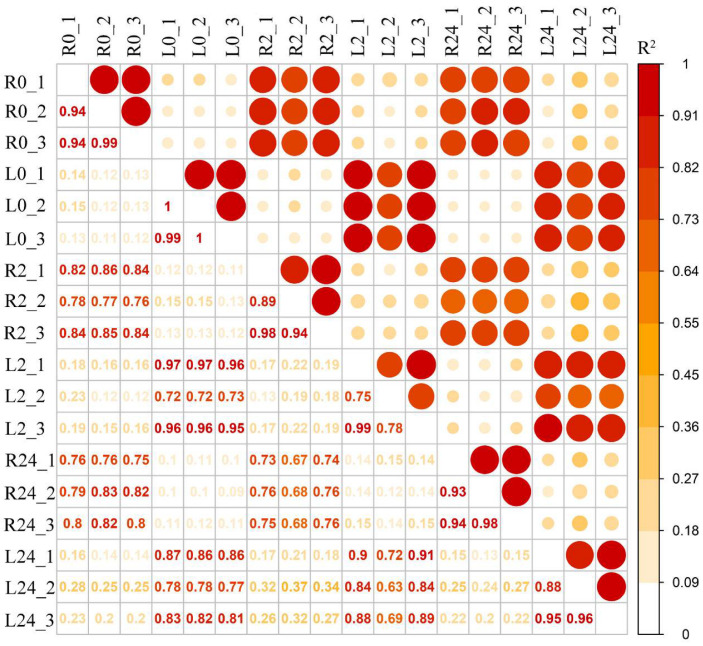
Pearson’s correlations between twelve samples. R^2^ represents the coefficient of determination. An orange background represents a greater coefficient of determination (L0: leaf 0 h, R0: root 0 h, L2: leaf 2 h, R2: root 2 h, L24: leaf 24 h, R24: root 24 h).

**Figure 3 ijms-23-08477-f003:**
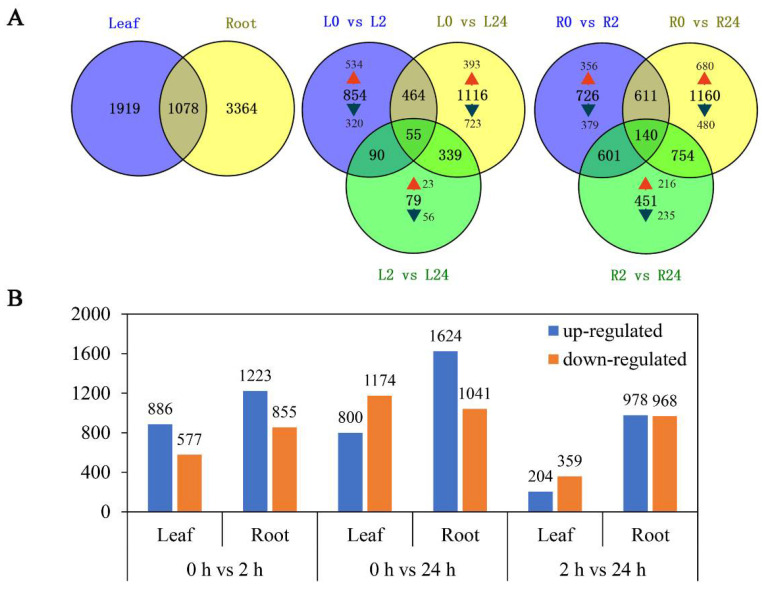
Summary of differentially expressed genes. (**A**) Summary of the numbers of root and leaf differentially expressed genes (DEGS) at different duration of salinity treatment. (**B**) Number of genes whose expression is differentially regulated between the different conditions. Orange bar: down-regulated genes; blue bar: up-regulated genes.

**Figure 4 ijms-23-08477-f004:**
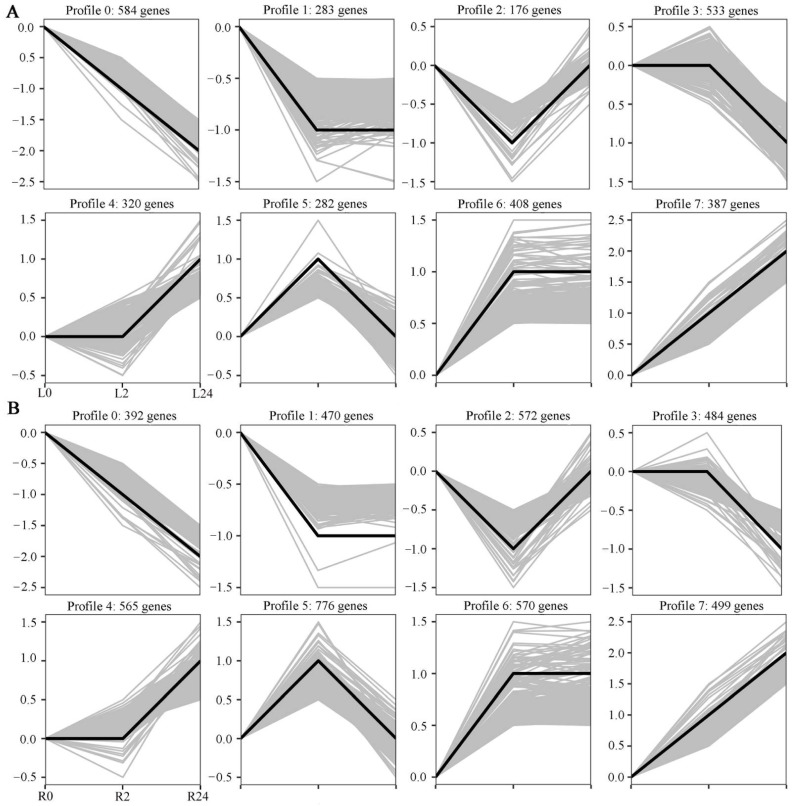
Differentially expressed gene expression patterns. The light grey lines represent the expression pattern of each gene, while the thick, dark lines represent the expression tendency of all the genes. (**A**) Patterns of gene expression in the leaves across three time points. (**B**) Patterns of gene expression in the roots across three time points.

**Figure 5 ijms-23-08477-f005:**
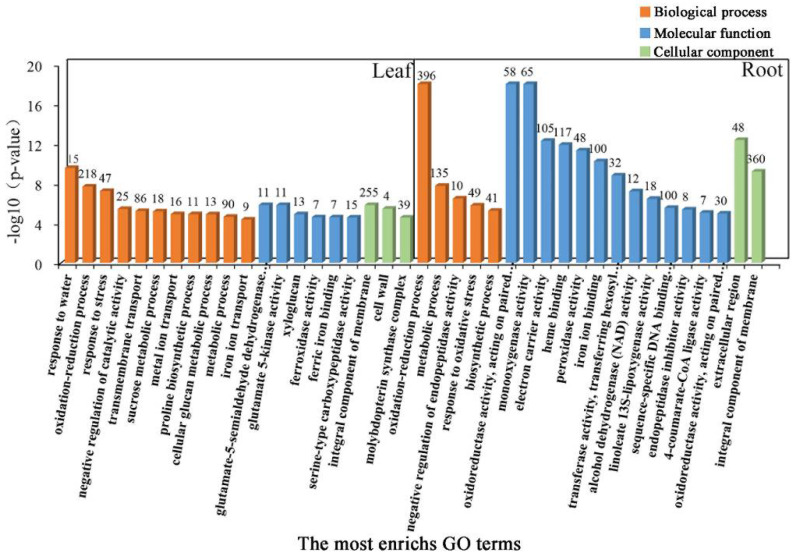
Gene Ontology (GO) enrichment analysis of DEGs. Genes were assigned to three main categories: Biological process, Molecular function, and Cellular component. The names of the GO categories are listed along the *x*-axis. The degree of GO enrichment is represented by the false discovery rate (FDR) value and the number of genes enriched in each category. The FDR value indicates the corrected *p*-value, ranging from 0 to 1, and an FDR value closer to 0 indicates greater enrichment.

**Figure 6 ijms-23-08477-f006:**
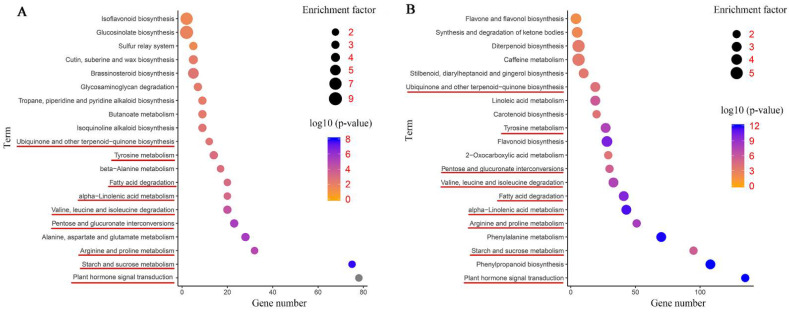
Kyoto Encyclopedia of Genes and Genomes (KEGG) pathway enrichment scatter diagram of DEGs. Only the top 20 most strongly represented pathways are displayed in the diagram. The degree of KEGG pathway enrichment is represented by an enrichment factor, the FDR value, and the number of genes enriched in a KEGG pathway. The enrichment factor indicates the ratio of differentially expressed genes enriched in this pathway to the total number of annotated unigenes in this pathway. The red underline represents the same enriched pathways among whole plants, leaves, roots, and co-induced DEGs. (**A**) KEGG enrichment analysis of the leaves. (**B**) KEGG enrichment analysis of the roots.

**Figure 7 ijms-23-08477-f007:**
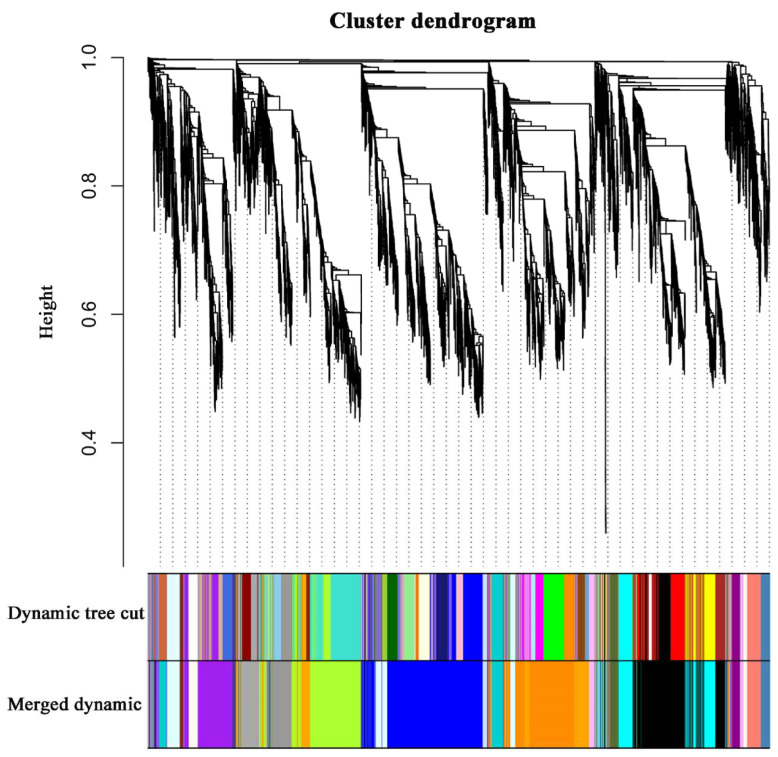
Clustering dendrograms of genes. Dissimilarity was based on topological overlap, together with assigned module colours. The 23 co-expression modules are shown in different colours.

**Figure 8 ijms-23-08477-f008:**
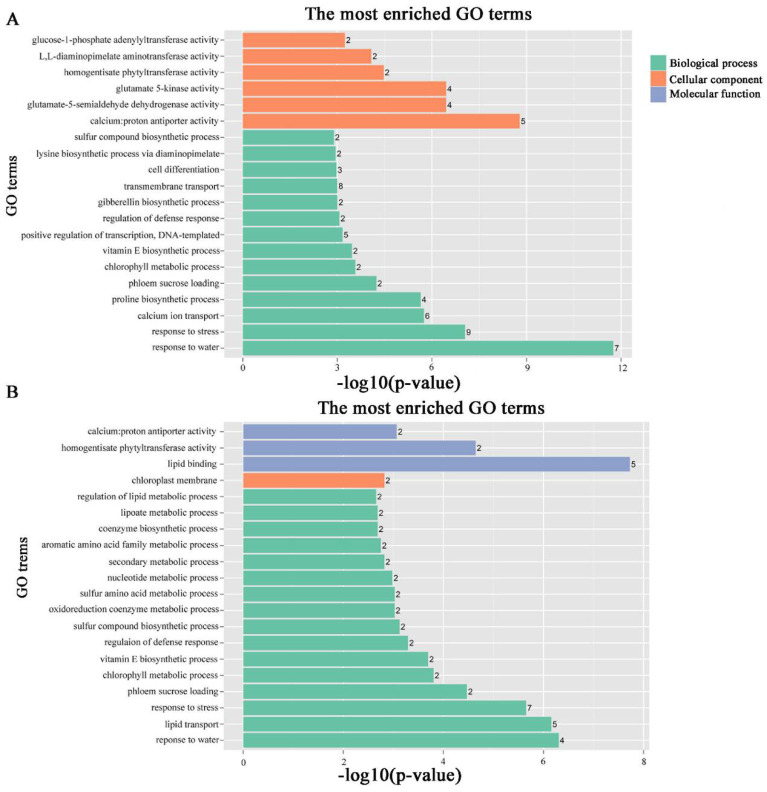
GO pathways associated with modules associated with stress-responsive traits. The number ahead of the items indicates the gene number. (**A**) Dark grey module GO pathways. (**B**) Purple module GO pathways.

**Figure 9 ijms-23-08477-f009:**
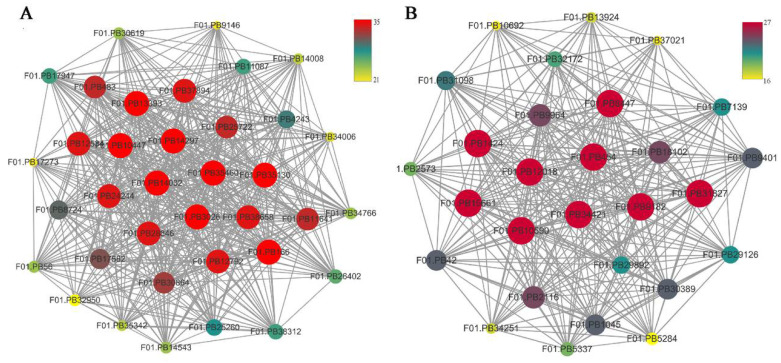
Network relationship among modules. (**A**) The thirty-six genes with the greatest MCODE score are in the dark grey module. (**B**) The twenty-eight genes with the greatest MCODE score are in purple module.

**Figure 10 ijms-23-08477-f010:**
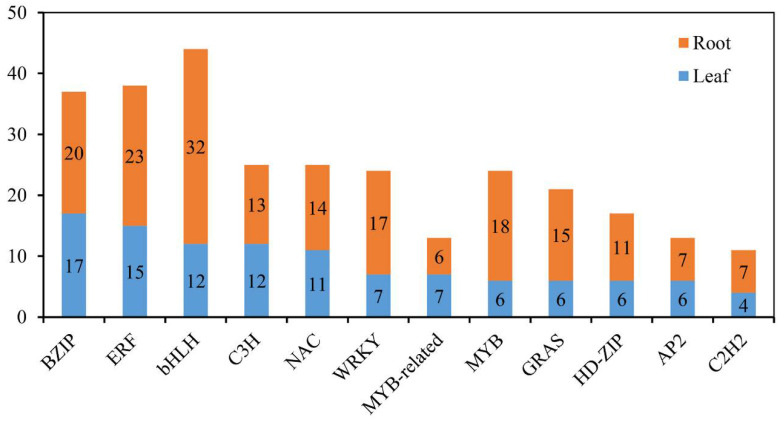
Distribution of the top 12 TFs responsive to salinity stress in common vetch.

**Figure 11 ijms-23-08477-f011:**
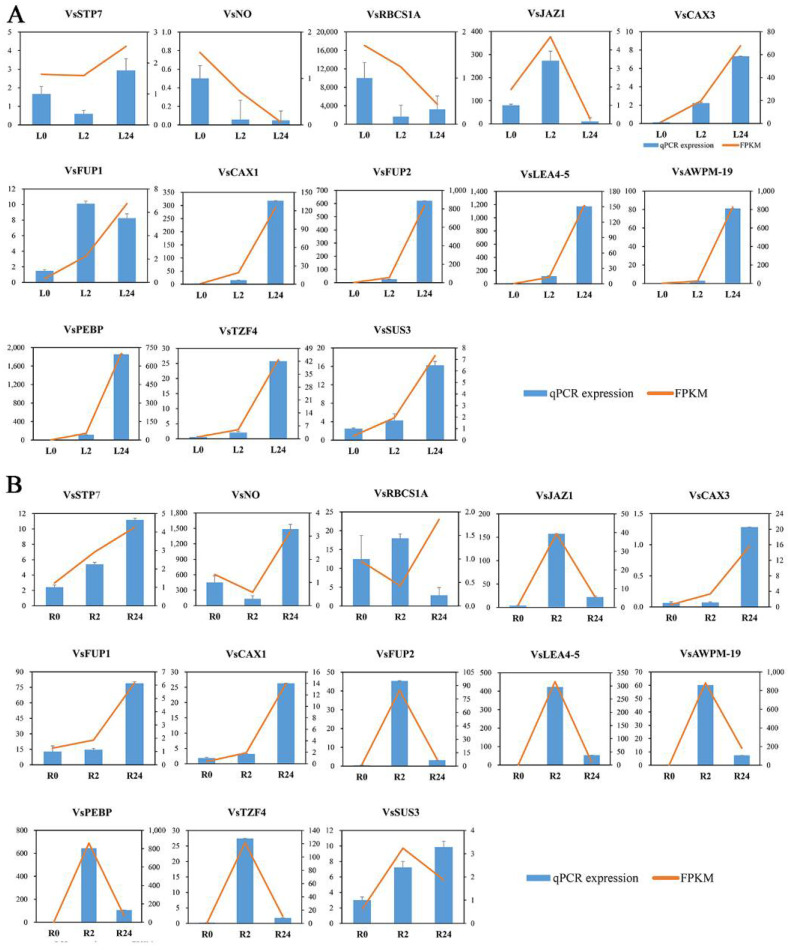
The expression patterns of thirteen selected genes identified by RNA-seq were verified by qRT-PCR. (**A**) Bar chart showing the expression changes in response to the L0 to L24 treatments for each candidate gene, as measured by RNA-seq and qRT-PCR. (**B**) Expression changes in response to the R0 to R24 treatments for each candidate gene, as measured by RNA-seq and qRT-PCR. STP7—sugar transporter protein 7; NO—NAD(P)-dependent oxidoreductase; RBCS1A—ribulose bisphosphate carboxylase small chain 1A; JAZ1—jasmonate-zim-domain protein 1; CAX3—cation exchanger 3; FUP1—function unknown protein 1; CAX1—cation exchanger 1; FUP2—function unknown protein 2; LEA4-5—late embryogenesis abundant 4–5; AWPM-19—ABA-induced wheat plasma membrane polypeptide-19; PEBP—phosphatidylethanolamine-binding protein; TZF4—CCCH-type zinc finger protein; SUS3—sucrose synthase 3.

**Figure 12 ijms-23-08477-f012:**
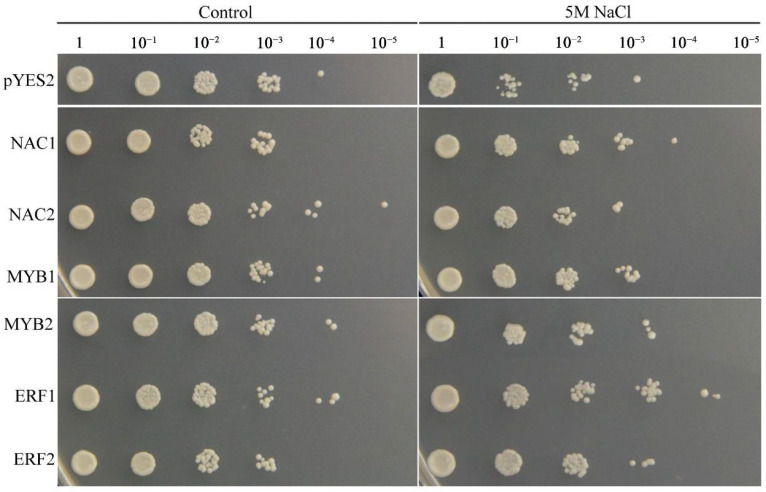
Phenotypic growth assays of *Saccharomyces cerevisiae* INVSc1 cells transformed with a pYES2 empty vector, *NAC1;2*, *ERF1;2*, or *MYB1;2* were spotted on SC-Ura media in 2 mL aliquots of 10-fold serially diluted (1, 10^−1^, 10^−2^, 10^−3^, 10^−4^, and 10^−5^) cultures under salinity stress.

**Table 1 ijms-23-08477-t001:** Illumina HiSeq high-throughput sequencing results and mapping ratio.

Sample Name	ReadSum	BaseSum	GC (%)	Q30 (%)	Total Mapped	Uniquely Mapped	Multiple Mapped
CL-0 h	25,651,280	7,672,360,004	42.65	92.32	79.42%	26.51%	73.49%
CL-0 h	27,015,618	8,079,601,436	42.69	92.62	79.72%	26.62%	73.38%
CL-0 h	28,011,278	8,374,751,502	42.77	91.78	79.45%	26.01%	73.99%
CR-0 h	26,924,164	8,045,472,396	42.93	90.68	76.02%	30.23%	69.77%
CR-0 h	26,568,062	7,930,048,480	42.69	90.77	76.75%	30.79%	69.21%
CR-0 h	25,307,189	7,551,301,734	42.79	90.35	76.09%	30.89%	69.11%
TL-2 h	25,859,885	7,713,984,870	42.81	92.18	80.22%	26.04%	73.96%
TL-2 h	23,973,183	7,167,050,540	45.33	91.2	75.45%	19.69%	80.31%
TL-2 h	28,540,428	8,532,717,458	42.88	91.19	79.23%	25.98%	74.02%
TR-2 h	34,472,027	1,030,8164,942	42.42	92.87	76.41%	31.41%	68.59%
TR-2 h	21,406,284	6,411,072,890	42.43	92.69	72.95%	32.61%	67.39%
TR-2 h	30,183,602	9,029,440,866	42.65	93.16	76.47%	30.74%	69.26%
TL-24 h	23,924,641	7,160,307,252	42.17	91.49	81.99%	26.76%	73.24%
TL-24 h	30,612,059	9,143,959,668	42.98	92.74	77.55%	28.48%	71.52%
TL-24 h	34,902,231	10,444,591,882	42.57	92.8	79.06%	28.08%	71.92%
TR-24 h	28,348,083	8,475,035,384	43.11	91.61	74.99%	30.10%	69.90%
TR-24 h	28,097,797	8,358,094,914	43.07	92.41	76.49%	29.67%	70.33%
TR-24 h	30,363,513	9,045,723,502	42.72	92.04	76.94%	29.67%	70.33%

## Data Availability

The raw sequencing data for 18 samples produced by the Illumina HiSeq X Ten platform can be accessed from the NCBI Sequence Read Archive (SRA) database (https://dataview.ncbi.nlm.nih.gov/object/PRJNA670057?reviewer=831uqd3isnhmifcti5ptk34ggn accessed on 1 October 2020) under accession numbers SRR12847380-SRR12847397, and the BioProject accession number is PRJNA670057.

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
