# Peer review of "Comparative Transcriptomic Analysis of Root and Leaf Transcript Profiles Reveals the Coordinated Mechanisms in Response to Salinity Stress in Common Vetch"

_ijms, 2022, doi:10.3390/ijms23158477_

Round 1

Reviewer 1 Report

This manuscript (MS) by Lin et al. includes genome-wide analysis of the response to salinity stress in leaves and roots of common vetch using RNA-seq data. This manuscript presents simple and interesting data that help us to understand the regulatory mechanism of the salinity stress response in common vetch. However, it should be revised before the publication in IJMS as follows.

1. The Materials and Methods section should come after Discussion according to IJMS guideline. Accordingly, the number of references should be changed.

2. Figure 3 is needed to be revised for better understanding of data in a clear way. It is complicated and difficult to understand. Especially, the Vann diagrams of current figure 3A and 3B are very confusing.

(1) In Figure 3A and 3B, comparing the number of DEGs is recommended to be separated into up-regulated genes and down-regulated genes.

(2) In Figure 3A, L2 vs L24 (R2 vs R24) data are difficult to understand. Especially, how can the expression patterns of the 23 up-regulated genes be explained? What is the argument of the L2 vs L24 data? I think that authors can argue their points without the data. The L2 vs L24 (R2 vs R24) data are recommended to be removed.

3. The meaning of Figure 12 should be discussed. Actually, it is difficult to understand the meaning of Figure 12. It seems that yeast and plants have basically different mechanisms in the salinity response in many aspects. I wonder if yeast can be used to check the function of plant transcription factors in salinity response. Actually, heterologous plant system (e.g. Arabidopsis), not yeast system, is recommended to analyze the function of the selected transcription factors. Authors need to explain and discussion the meaning of Figure 12 (or remove it).

4. In Discussion, some statements are necessary to add the citations of references or results.

5. Discussion is too long. It needs to be shortened.

Reviewer 2 Report

Dear Authors, 

the manuscript was very well prepared, I find no significant factual errors. The work is quite extensive and has a lot of results. Sometimes the captions in the figures are not entirely clear, these are abbreviations that should be explained, sometimes they should be grouped, identified, e.g. in figure 2. The titles should also contain full names and abbreviations in parentheses e.g. chapter 3.5. My main remark is that the Authors of the work take note of the descriptions in the figures and correct them so that the reader would find them clear, legible and understandable, e.g. by developing abbreviations in figure captions.

Round 2

Reviewer 1 Report

This manuscript (MS) by Lin et al. is properly revised based on reviewer’s comments. However, a few issues need to be solved before the publication in IJMS as follows.

1. Figure 3 is needed to be revised for better understanding of data in a clear way. Especially, in Figure 3A and 3B, comparing the number of DEGs should be separated into up-regulated genes and down-regulated genes (or the information of the up-regulated genes and down-regulated genes should be included in the manuscript). The information of the up-regulated genes and down-regulated genes (as well as the DEG information) is important to understand which group genes (or genes) are up-regulated and down-regulated in the response to salinity stress in leaves and roots of common vetch. However, the detailed information of the up-regulated genes and down-regulated genes is not clearly found in current manuscript.

2. Authors need to check the position of Materials and Methods section in the manuscript (after or before Conclusion) according to IJMS guideline.

Round 3

Reviewer 1 Report

The manuscript (MS: ijms-1799345) is properly revised according to reviewer’ comments. I think that this MS can be accepted for the publication in IJMS in current form.